# Polydextrose with and without *Bifidobacterium animalis* ssp. *lactis* 420 drives the prevalence of *Akkermansia* and improves liver health in a multi-compartmental obesogenic mice study

Christian Clement Yde [1,2]*, Henrik Max Jensen[1], Niels Christensen[1], Florence Servant[3], Benjamin Lelouvier[3], Sampo Lahtinen[4], Lotta K. Stenman[4], Kaisa Airaksinen[4], Henna-Maria Kailanto[4]

1 IFF Enabling Technologies, Brabrand, Aarhus, Denmark, 2 Department of Food Science, Aarhus University, Aarhus N, Denmark, 3 Vaiomer, Labège, France, 4 IFF Health & Biosciences, Kantvik, Finland

* Christian-Clement.Yde@iff.com

## Abstract

The past two decades of research have raised gut microbiota composition as a contributing factor to the development of obesity, and higher abundance of certain bacterial species has been linked to the lean phenotype, such as *Akkermansia muciniphila*. The ability of pre- and probiotics to affect metabolic health could be via microbial community alterations and subsequently changes in metabolite profiles, modulating for example host energy balance via complex signaling pathways. The aim of this mice study was to determine how administration of a prebiotic fiber, polydextrose (PDX) and a probiotic *Bifidobacterium animalis* ssp. *lactis* 420 (B420), during high fat diet (HFD; 60 kcal% fat) affects microbiota composition in the gastrointestinal tract and adipose tissue, and metabolite levels in gut and liver. In this study C57Bl/6J mice (N = 200) were split in five treatments and daily gavaged: 1) Normal control (NC); 2) HFD; 3) HFD + PDX; 4) HFD + B420 or 5) HFD + PDX + B420 (HFD+S). At six weeks of treatment intraperitoneal glucose-tolerance test (IPGTT) was performed, and feces were collected at weeks 0, 3, 6 and 9. At end of the intervention, ileum and colon mucosa, adipose tissue and liver samples were collected. The microbiota composition in fecal, ileum, colon and adipose tissue was analyzed using 16S rDNA sequencing, fecal and liver metabolomics were performed by nuclear magnetic resonance (NMR) spectroscopy. It was found that HFD+PDX intervention reduced body weight gain and hepatic fat compared to HFD. Sequencing the mice adipose tissue (MAT) identified *Akkermansia* and its prevalence was increased in HFD+S group. Furthermore, by the inclusion of PDX, fecal, ileum and colon levels of *Akkermansia* were increased and liver health was improved as the detoxification capacity and levels of methyl-donors were increased. These new results demonstrate how PDX and B420 can affect the interactions between gut, liver and adipose tissue.

**Data Availability Statement:** All relevant data are within the manuscript and its Supporting Information files.

**Funding:** This study was funded by IFF Health & Biosciences. CCY was financially supported by the Innovation Fund Denmark (Project No. 4228-00010B). IFF Health & Biosciences manufactures the prebiotic (Litesse® polydextrose) and probiotic (HOWARU® Shape (10B CFU B. lactis B420™) used in this study. Investigating the effects of these commercial products was the reason to initiate the study incl. study design, data collection and analysis and decision to publish. The specific roles of these authors are articulated in the 'author contributions' section.

**Competing interests:** All authors have been affiliated at either IFF or Vaiomer. IFF Health & Biosciences manufactures the prebiotic (Litesse® polydextrose) and probiotic (HOWARU® Shape (10B CFU B. lactis B420™) used in this study. Vaiomer is a contract research organization and has no competing interests. Vaiomer provided the animal study and further microbiome data analysis as specified in author contribution section. This does not alter our adherence to PLOS ONE policies on sharing data and materials.

**Abbreviations:** PDX, polydextrose; B420, *Bifidobacterium animalis* subsp. *lactis* 420; CPMG, Carr-Purcell-Meiboom-Gill sequence; GCMS, gas chromatography mass spectrometry; HFD, high fat diet; MAT, mesenteric adipose tissue; LCMS, liquid chromatography mass spectrometry; LDA, Linear discriminant analysis; NC, normal control; NMR, nuclear magnetic resonance; Noesy, nuclear overhauser spectroscopy); OMV, outer membrane vesicles; OTU, Operational taxonomic units; PCA, principal component analysis; PCOA, principal coordinate analysis.

# Introduction

Despite an increased awareness of obesity having a negative impact on human health, according to OECD almost one out of four adults in OECD countries (24%) is obese [1]. In addition to exercise and diet, the past two decades of research have shown many other factors, including gut microbiota composition, to contribute to the development of obesity [2, 3]. In 2014 an expert committee defined probiotics as live microorganisms that, when administered in adequate amounts, confer a health benefit on the host [4]. Also, a consensus has been reached to state that prebiotics are substrates selectively used by the host microbiota conferring a health benefit [5]. The microbiota can influence the host through several mechanisms including energy balance and appetite regulations. As reviewed by Kobyliak and coworkers [6], intervention studies in both animals and humans demonstrate that the intake of probiotics can assist in reducing body weight gain in animals on a high energy diet and ameliorate metabolic health risk factors such as insulin resistance in humans. A probiotic strain *Bifidobacterium animalis* ssp. *lactis* 420 (B420) has shown potential as weight controlling agent, since administrating it has resulted in a reduction of body fat mass accumulation in mice fed the high fat diet (HFD) [7, 8]. A synergistic effect between B420 and a prebiotic soluble fiber, polydextrose (PDX) [9, 10], has been indicated by our group, when a lower accumulation of body fat mass with the combination of these two ingredients was noted in humans during a 6-month intervention period in a randomized double-blind clinical trial [11]. PDX, which is a non-digestible oligosaccharide consisting of randomly-bonded glucose units, has reported prebiotic properties including partial fermentation by colonic bacteria, SCFA production and modification of fecal microbiota [12–14]. PDX has also shown potential in supporting metabolic health and weight management, as reported in several preclinical and clinical studies [15–18].

Alteration of gut microbiota is known to impact human health in a growing number of health areas. Several publications have shown correlations between obesity and diet, probiotic and prebiotics interventions in relation to changes in fecal levels of short chain fatty acids (SCFA) [19], amino acids [20, 21] and biogenic amines [22]. In a paper by Zhao *et al.* [23] deficiency in SCFA production from carbohydrate fermentation by specific strains was linked to type-2 diabetes. A multi-compartment approach enables the analysis of gut-liver axis functions, as metabolite levels influenced the bidirectional communication between gut and liver [24]. However, the precise mode of action and molecular mechanisms behind these fascinating links are not understood. Metabolomics and metagenomics are both techniques that together with a multi-compartmental approach can provide a mechanistic understanding of the underlying mechanisms of intake of pro- and prebiotics [25, 26]. The strength of combining the study of gut microbiota and analysis of small-molecule metabolites in relation to human physiology was recently reviewed [27].

Certain bacterial species have been identified to be present in higher abundance in persons with lean phenotype, and others with obesity. Species that belong to genus *Akkermansia*, such as *Akkermansia muciniphila*, have been getting a lot of attention during the past years as they have increasingly shown to be associated with lean phenotype and good metabolic health in earlier studies [28]. A proof-of-concept exploratory clinical trial with supplemented *A. muciniphila* has been conducted recently showing improved insulin sensitivity and reduced insulinemia and plasma total cholesterol [29]. Furthermore, the oral administration of *A. muciniphila* has been shown to reverse HFD-induced metabolic disorders via increased mucin layer thickness, to reduce endotoxemia, and to even increase the number of goblet cells in the epithelium [30]. In a recent study, the beneficial effects of *A. muciniphila* were recapitalized partially by a membrane protein called Amuc_1100 isolated from the outer membrane of the bacterium via its interaction with Toll-like receptor 2 [31]. In a recent randomized clinical trial B420 and

PDX were shown to induce compositional shift in human microbiota including increased prevalence of *Akkermansia* [25]. Thus, it is of great interest to better understand how B420 and PDX administration during HFD affects mice microbiota composition not only in various parts of the gastrointestinal tract, but also in adipose tissue, as well as the metabolites in gut and tissues.

In the present multicompartmental study, the microbiota composition in feces, ileum, colon and adipose tissue was analyzed using 16S rDNA sequencing and NMR metabolomics on fecal extracts using solution NMR and intact liver tissue using high resolution magic angle spinning (HR-MAS) NMR spectroscopy [32, 33]. The aim of the present study was to examine the underlying mechanisms of PDX and B420 treatments explaining the earlier findings in animal studies and human clinical trials on metabolic health. Specifically, our aim was to study the cross-talk between gut and liver as demonstrated by the observed changes in the microbiota composition as well as bacterial and host metabolite levels induced by the HFD and probiotic, prebiotic or synbiotic intervention using a C57BL/6J mice model.

## Materials and methods

### Animals and study design

All animals were male C57Bl/6J mice obtained from Charles River (L'Arbresle, France) and acclimatized for at least seven days prior to any experimentation. Mice were housed in groups of five animals per cage and maintained under a normal dark-light cycle (12 h /12 h), 22±2°C and 55±10% relative humidity. Tap water and feed were provided ad libitum. At ten weeks of age, intervention week 0, each group of mice started one of the five treatments and gavaged daily (n = 40 per arm): 1) Normal control (NC; RM1-Special Diets Service); 2) High-fat diet (HFD, 60 kcal% fat, 20 kcal% protein, 20 kcal% carbohydrate, 5.21 kcal/kg; Research Diets Inc. D12492); 3) HFD + polydextrose (HFD+PDX, 60 kcal% fat, 20 kcal% protein, 20 kcal% carbohydrate, 5.21 kcal/kg, Litesse® Ultra™ (Danisco USA Inc., Terre Haute, IN, USA) 200 mg/day); 4) HFD + *Bifidobacterium animalis* ssp. *lactis* 420 (HFD +B420, 60 kcal% fat, 20 kcal% protein, 20 kcal% carbohydrate, 5.21 kcal/kg, $10^9$ CFU/day), or 5) HFD (60 kcal% fat) + PDX (200 mg/day) + B420 ($10^9$ CFU/day) also referred to as HFD+S. Two experiments were performed, where experiment I included 10 mice per group and experiment II included 30 mice per group. One mouse died at intervention week 7 prior to completion of the study. The clinical data and metagenomics data were combined for both experiments I and II, whereas the metabolomics data were split into each experiment due to experiment-wise variation between experiments I and II in these analyses. Body weight was recorded every week from the age of 10 weeks until the age of 19 weeks week (intervention week 0 to 9). Mice studies were performed at the French biotech company and contract research organization, Vaiomer, in accordance with the article R-214-89 of the French "Code rural et de la pêche maritime" section 6 "Use of living animals for scientific research" and approved by the ethical committee CEEA-122 of the SICOVAL Prologue Biotech Institute.

### Glucose tolerance tests

An intraperitoneal glucose-tolerance test (IPGTT) was performed after six weeks of treatment to assess glucose management in mice. Six-hour-fasted mice were injected with glucose (1 g/kg) into the peritoneal cavity. The glucose response was followed from 30 min before the glucose challenge until 120 min after the challenge, measuring plasma glucose every 15–30 min using a standard glucose meter (Roche Diagnostics, Basel, Switzerland).

## Sample collection and processing

Feces were collected fresh (between 9am and 11am) at intervention weeks 0, 3, 6 and 8 and stored at -80˚C until analyzed. After the end of the intervention, intervention week 9, the mice were anesthetized in the fasting state (overnight). Ileum and colon mucosa were collected after a quick removal of the content, the mesenteric adipose tissue was collected (without the mesenteric lymph nodes). All samples were stored in plastic tube containing TRIzol (Invitrogen) and frozen in liquid nitrogen. Liver was harvested and frozen in liquid nitrogen. All samples were stored at -80˚C.

## DNA isolation and microbiota sequencing

Total DNA was extracted from frozen samples using the TRIzol protocol according to the manufacturer's instructions and as previously described [34]. Protocols for PCR and sequencing were similar to the methods performed in Hibberd *et al.* [25] paper with the following modification for the intestinal and adipose tissues due to the low bacterial DNA template: the Taq used was instead Platinum Taq DNA Polymerase (ThermoFisher Scientific, Waltham, MA) with 35 PCR cycles (rather than 30) and an increase to 30% PhiX (rather than 15%) spike-in for the Illumina MiSeq sequencing run. 16S rDNA bacterial profiles associated with 780 fecal, 198 ileum, 202 colon, and 197 mesenteric adipose tissue samples from the mice were collected. The targeted metagenomic sequences from microbiota were analyzed using the bioinformatics pipeline established by Vaiomer from the FROGS guideline (described in detail in an [35]). Briefly, after demultiplexing of the bar-coded Illumina paired reads, single read sequences were cleaned and paired for each sample independently, into longer fragments. OTUs were produced with via single-linkage clustering and taxonomic assignment was performed to determine community profiles. As a software to trim sequences, fastx trimmer from FASTX Toolkit v0.0.14 was used with the following parameters: R1 and R2 were cut to 200nt due to the amplicon size (around 250nt with the primers used). As bioinformatics pipeline, FROGS v1.3.0 was used with following filters: Amplicons with a length < 200nt or a length > 300nt were removed; Amplicons with at least one ambiguous nucleotides ('N') were removed; OTU identified as chimera (with vsearch v1.9.5) in all samples in which they were present were removed; OTU with an abundance lower than 0.005% of the whole dataset abundance were removed; OTU with a strong similarity (coverage and identity > = 80%) with the phiX (library used as a control for Illumina sequencing runs) were removed. The clustering was produced in two passes of the swarm algorithm v2.1.6. The first pass was a clustering with an aggregation distance equal to 1, and the second pass was a clustering with an aggregation distance equal to 3. The taxonomic assignment was produced by Blast+ v2.2.30+ with the databank RDP v11.4. The R package PhyloSeq v1.14.0 was used to perform the microbiome diversity analyses and graphics. The samples with less than 5000 sequences after FROGS processing were not included in the statistics.

## NMR spectroscopy

The NMR measurements of the fecal extracts were performed on a 600 MHz Bruker Avance III spectrometer and the intact liver were performed on a 600 MHz Bruker Avance spectrometer both operating at a frequency of 600.13 MHz for $^{1}H$ nucleus (Bruker Biospins, Rheinstetten, Germany).

Fecal samples were extracted in a 1:4 weight-to-buffer ratio using a 0.75 M phosphate-buffered saline solution (pH = 7.4). The samples were homogenized by whirl mixing for 2 min and centrifuged at 10,000g for 15 min at 4˚C. A volume of 500 μl supernatant was mixed with 100 μl of $D_2O$ containing 0.05% 3-(Trimethylsilyl)propionic acid-d$_4$ sodium salt (TSP) as an

internal standard (IS). A standard 1D Noesy experiment with pre-saturation (Bruker "noe-sypr1d" sequence) was used to acquire $^1$H NMR spectra at 298 K using a 5-mm broadband observe (BBO) probe. A total of 128 scans collected into 64K data points were acquired with a spectral width of 11.97 ppm, a recycle delay of 3 s and an acquisition time of 4.55 s. The fecal $^1$H spectra were processed with an exponential line-broadening of 0.8 Hz prior to the Fourier transformation. Automatic metabolic deconvolution and quantifications of 33 metabolites were performed using the Bayesian AuTomated Metabolite Analyser (BATMAN) R package [36].

For the intact liver High Resolution Magic Angle Spinning (HR-MAS) analysis, a piece of each of the intact liver samples (still frozen) were packed at -20˚C in disposable pre-weighed 50 µL inserts (Bruker Biospin, Rheinstetten, Germany) followed by addition of 10 µL of $D_2O$ containing 0.05% TSP. Upon measurement, the insert (sample) was placed in a 4 mm zirco-nium rotor (Bruker BioSpin) and $^1$H NMR spectra were acquired with a CPMG experiment (cpmgpr1d, Bruker sequence) using a 4 mm HR-MAS probe (Bruker BioSpin). The acquisition parameters for the spectra were as follows: 5 kHz spin rate, 64 scans, a spectral width of 17.36 ppm with 32K data points, a total spin−spin relaxation delay of 100 ms (2nτ), a spin −echo delay of 1 ms (τ), a recycle time of 3 s and an acquisition time of 1.57 s. An exponential line broadening function of 0.3 Hz was applied to the free induction decay prior to the Fourier transformation. The Chenomx NMR Suite 8.4 software was applied to assign and quantify metabolites by determining the relative area of each metabolite and normalization to the mass of each liver sample. The methyl lipid signal at 1.33 ppm was manually integrated as a mea-surement of the relative hepatic lipid content and normalized to the mass of each liver sample.

Each spectrum was automatically phased and referenced to the anomeric signal of a-glucose at 5.23 ppm for intact liver samples and TSP at 0.0 ppm for feces samples using TopSpin 3.5 (Bruker BioSpin).

## Data analysis

All clinical data were analyzed by one-way ANOVA with univariate information carrier com-prising area under curve for IPGTT and log-transformed growth rate (slope) for body weight growth. If the global P-value was significant, Tukey's multiple comparisons test was used to assess differences between groups. All data are expressed as mean ± standard error of mean (SEM), and significances are two-sided. Differences were considered statistically significant when $P < 0.05$.

Residual water signal and spectral ends were removed from the $^1$H NMR spectra. The intact liver samples were normalized to sample weight. For each sample type, the NMR spectra were imported into Matlab 2017a (The MathWorks, Inc., Natick, MA, USA), baseline-corrected by distribution-based classification [37], aligned using icoshift version 1.3.1 [38] and binned according to an optimized bucketing algorithm [39]. In addition, multivariate data analysis was performed by PCA analysis on Pareto-scaled data using PLS Toolbox (eigenvector Research, U.S.A.) in Matlab R2016b to detect clustering behavior. Univariate statistics using mixed models in SAS statistical software package version 9.3 (SAS Institute, Cary, NC, USA) were performed on the quantified fecal metabolites with treatment and week effects and their interaction. The NMR variables were log-normal distributed and baseline-adjusted with week 0 samples prior to modelling. The effects of treatment and experiment on the quantified liver metabolites were analyzed using a two-way ANOVA in R (v. 3.5.1).

LDA Effect Size (LEfSe) [40] is an algorithm for high-dimensional biomarker discovery and explanation that can identify taxonomic groups characterizing the differences between two or more biological conditions. LEfSe first robustly identified features that were statistically different

among biological classes. It then performed additional tests to assess whether these differences are consistent with respect to expected biological behavior. The OTU files were uploaded and formatted for LEfSe analysis using the per-sample normalization of sum values option. The linear discriminant analysis effect size was determined using default values (alpha value of 0.5 for both the factorial Kruskal-Wallis test among classes) except for the logarithmic LDA score threshold for discriminative features which was set to 4.3. The strategy for multi-class analysis was set to 'all-against-all'. Differential features detected as biomarkers from the raw data used to generate the cladograms were plotted as abundance histograms with class information. The biomarkers found by LEfSe were ranked according to their effect size and associated by color (red and green for pairwise analysis) with the class (group) having the highest median.

Spearman's rank correlation analysis adjusted for differences between experiments I and II was performed on body weight gain, fecal and intact liver metabolites and relative abundance of *Akkermansia* using SAS (v. 9.3). Fisher-Z transformation was determined for experiments I and II. Afterwards, a single correlation for each treatment was calculated by the inverse Fisher-Z transformation to the standard weighted average of experiments I and II Z-scores. Two-sided P values for each correlation were also included. Visualization by heat maps were done using R (v. 3.5.1) and gglot2 package (R Development Core Team, 2008).

## Results

### Body weight development and glucose tolerance as an effect of B420 and PDX

The C57BL/6J mouse model with the intake of HFD showed significant (NC vs. HFD: p<0.001) increased body weight gain during the intervention from weeks 0 to 9 (age weeks 10 to 19) when compared to a NC diet (Fig 1A). Treatment HFD+PDX (HFD *vs*. HFD+PDX, p = 0.020) also reduced body weight gain during 8 weeks of intervention compared to HFD alone. In addition, HFD impaired glucose tolerance for all HFD groups and no effect of pre-, pro- and synbiotics were observed (Fig 1B).

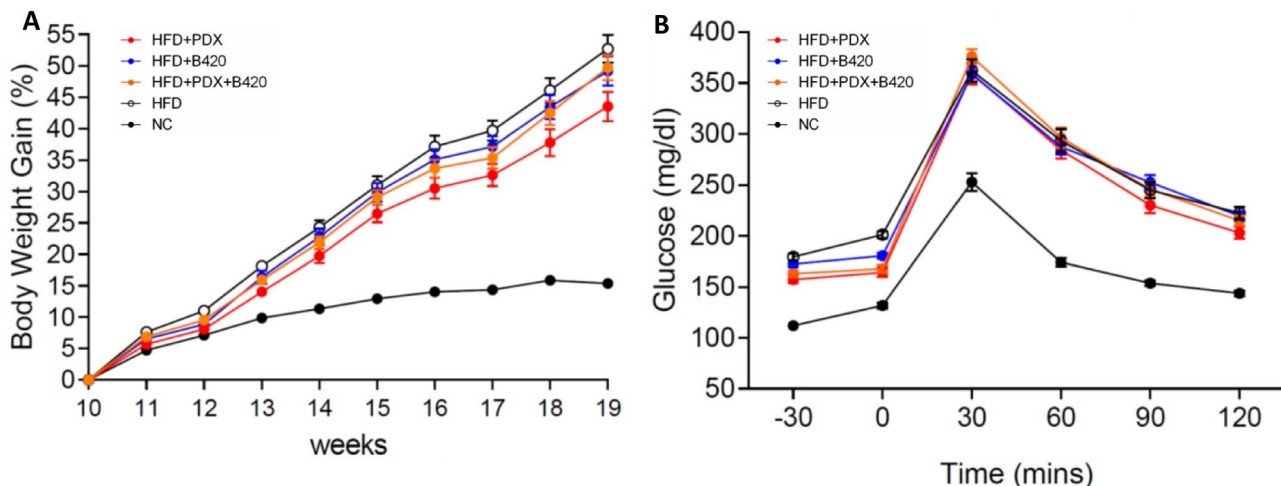

**Fig 1. Treatment effects in male C57Bl/6J mice during 19-weeks intervention for 1st and 2nd experiment.** Values are means ± SEM. (A) Body weight gain (%). Significance: NC *vs*. all HFD, p<0.001; HFD *vs*. HFD+B420, p = 0.959; HFD *vs*. HFD+PDX, p = 0.020; HFD *vs*. HFD+S, p = 0.712; HFD+B420 vs. HFD+PDX, p = 0.127; HFD+B420 vs. HFD+PDX+B420, p = 0.979; HFD+PDX vs. HFD+PDX+B420, p = 0.366. (B) intraperitoneal glucose-tolerance test (mg/dl). Significance: NC vs. all HFD, p<0.001; HFD vs. HFD+B420, p = 0.987; HFD vs. HFD+PDX, p = 0.214; HFD vs. HFD+S, p = 0.975; HFD+B420 vs. HFD+PDX, p = 0.483; HFD+B420 vs. HFD+PDX+B420, p = 1.000; HFD+PDX vs. HFD+PDX+B420, p = 0.545.

## Microbiota diversity and clustering of intervention groups

Looking at the relative abundance at Genus level revealed a similar microbial composition pattern between HFD and HFD+B420 groups and between HFD+PDX and HFD+S groups for the fecal weeks 3-6-8 and ileum and colon samples at week 9 (S1 Fig). No difference was observed between the groups for fecal samples at baseline and adipose tissue (MAT) (S1A and S1G Fig), indicating that there were no pre-treatment differences between the groups. From week 3 and onwards, a clear pattern was seen in the fecal samples with increased abundance of *Alistipes* and *Bacteroides* in all HFD groups. PDX also increased the relative abundance of *Akkermansia* and *Parabactoroides* (S1B–S1D Fig). In ileum and colon HFD and HFD+B420 did not induce clear systematic changes in the relative abundance at Genus level, whereas PDX induced increased relative abundance of *Akkermansia* and additional for most mice *Parabacteroides* in both the HFD+PDX and HFD+S groups.

Shannon index was used to represent α-diversity (within-sample species diversity), and the α-diversity was significantly decreased for all HFD groups in all fecal samples at week 8 and colon samples (S2B and S2D Fig). β-diversity was visualized using PCoA and the Bray-Curtis method. No differences in β-diversity were found at fecal baseline between the groups (S3A Fig). After intervention, PCoA plots of fecal, ileum, colon and MAT samples showed separation between NC and HFD groups and distinct clusters for PDX in the diet (S3B–S3E Fig).

Furthermore, the pairwise group LEfSe analysis (LDS >4.3) of the microbiota in fecal samples showed no difference between groups at baseline (Fig 2A). On a Phylum level, HFD increased the prevalence of *Firmicutes*, and decreased that of *Bacteroidetes*, in 8-week fecal, ileum and colon samples (Fig 2B–2D). On a genus level, in fecal microbiota, *Alistipes* and *Bacteroides* were increased in HFD when compared to NC, and with the inclusion of PDX the fecal relative abundance of *Akkermansia* and *Parabacteroides* were increased (Fig 2B). In the gut, relative abundance of *Barnesiella* was decreased in the ileum (Fig 2C) and *Bacteroides* and

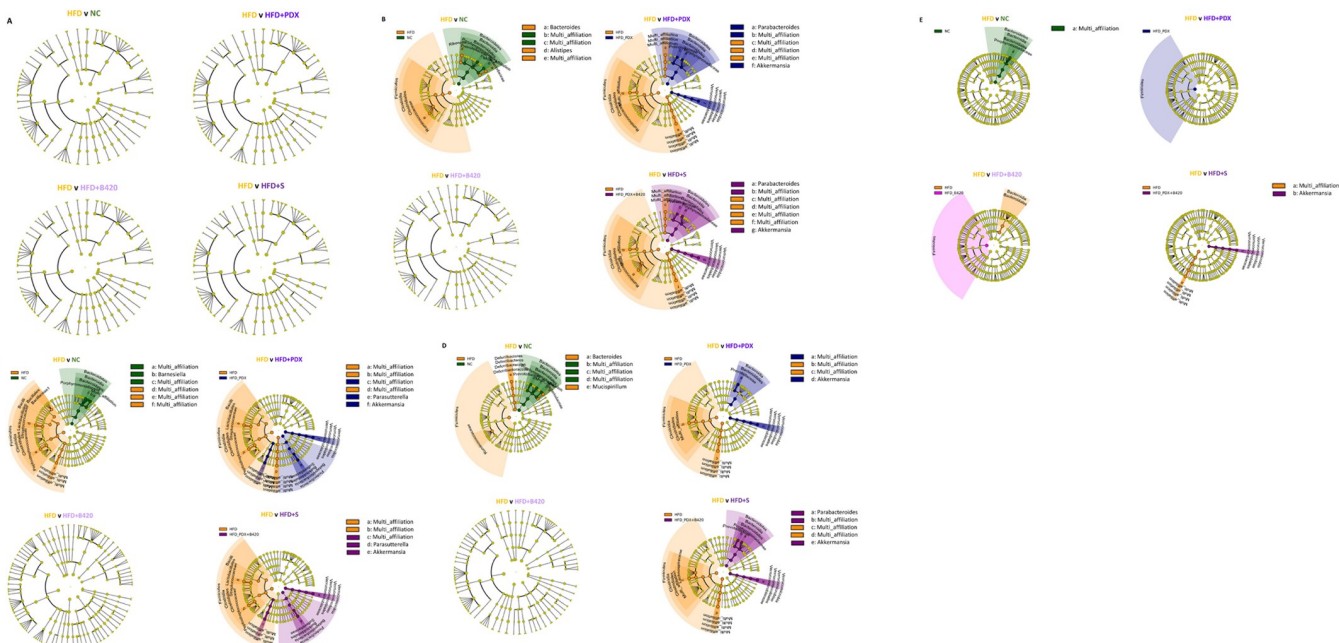

**Fig 2. Cladogram shown for LEfSe analysis performed on the complete sequence data.** The significant LDA scores for the bacterial taxon found by LEfSe were ranked according to their effect size and associated by color with each treatment. (A) Fecal samples at week 0. (B) Fecal samples at week 8. (C) Ileum samples. (D) Colon samples. (E) Adipose tissue (MAT) samples.

*Mucispirillum* were decreased in the colon (Fig 2D) in the HFD groups in comparison to NC. In addition, the relative abundance of *Akkermansia* increased in the ileum and colon as an effect of PDX intake, the relative abundance of *Parasutterella* increased in the ileum as an effect of PDX intake while the relative abundance of *Parabacteroides* increased in the synbiotic HFD+S group (Fig 2B–2D). B420 supplementation alone did not modify the observed changes on a Genus level for fecal, colon or ileum (Fig 2B–2D). On the contrary, in the adipose tissue (MAT), the abundance of *Akkermansia* was increased with the synbiotic treatment only (Fig 2E).

## The effect of pre- and probiotic on the metabolome

To get an overview of the effect of intake of an HFD diet together with probiotic, prebiotic or synbiotic interventions on the metabolome, PCA models for metabolomics data obtained from each compartment and sample type were performed. In feces, NC mice and fecal week 0 samples clustered together and the inclusion of PDX in the HFD diet, either alone or in synbiotic product, separated the HFD mice into two clusters (Fig 3A). From weeks 3 to 8, no changes in the fecal metabolome were seen. To obtain a better model of the changes in fecal metabolites without the dominating PDX signal, a targeted approach by automatic metabolic deconvolution was performed (Table 1). The baseline-adjusted fecal metabolic changes showed that HFD alone decreased the fecal SCFA levels of acetate, formate and propionate (P<0.001). The HFD +B420, HFD+PDX and HFD+S groups all had fecal SCFA levels closer to or similar to those observed in the NC group. BCFA (2-methyl butyrate and 3-methyl-2-oxovalerate) was decreased in the NC group, whereas it was increased in all HFD groups during the intervention (P<0.0001). Glycerol was increased the most with the inclusion of PDX (P<0.0001). In addition, ethanol levels increased in all groups, but the increase was highest in the HFD and HFD +B420 groups (P<0.0001). Overall, the concentrations of the majority of the measured fecal amino acids were increased most in the HFD+B420 group and least in the NC group. The changes of the TCA cycle intermediates including fumarate, malate and succinate in the fecal samples during the intervention were increased more in both HFD+PDX and HFD+S groups than in HFD group (all P<0.0001). The fecal glucose levels increased in the NC group and decreased in HFD and HFD+B420 groups during intervention, whereas inclusion of PDX resulted in no change in the synbiotic group and a slight decrease was observed in the HFD +PDX group (P<0.0001). The concentration of trimethylamine (TMA) in feces was found to be elevated after intervention with B420 alone and in combination with PDX (P<0.0001).

For the intact liver samples in experiment I, all the NC samples had negative scores in PC1 (62.6% explained variation), which corresponded to a decreased hepatic liver fat (Fig 3B). Although the different treatment groups were otherwise heavily overlapped in the PCA scores plot making it difficult to observe clustering, they showed some clustering in experiment II with mice consuming B420 in the upper-right quadrant, mice consuming PDX in the upper-left quadrant, mice solely on HFD in the right quadrants, and the synbiotic group with NC mice in the lower-left quadrant. These differences were mainly due to higher hepatic fat levels observed in HFD and HFD+B420 mice than in NC, HFD+PDX and HFD+S mice, which was in agreement with the integrated methyl lipid signal reported in Table 2 (P<0.0001). Furthermore, Table 2 shows that acetate (P<0.0001), ethanol (P = 0.001) and glycerol (P<0.0001) levels were elevated in NC mice and with the inclusion of PDX compared to HFD and HFD+B420 mice. For the quantified amino acids, the pattern was not that clear except that there seemed to be an increase in some of the amino acids in the liver in HFD+PDX and HFD+S groups. Both hepatic betaine and choline levels were significantly higher in HFD mice with the inclusion of PDX alone or together with B420 (P<0.0001). The energy-related metabolites creatine (P<0.0001), glucose (P = 0.0008), glycogen (P = 0.046) and lactate (P = 0.0002) all had higher concentrations in the HFD+PDX and HFD+S groups

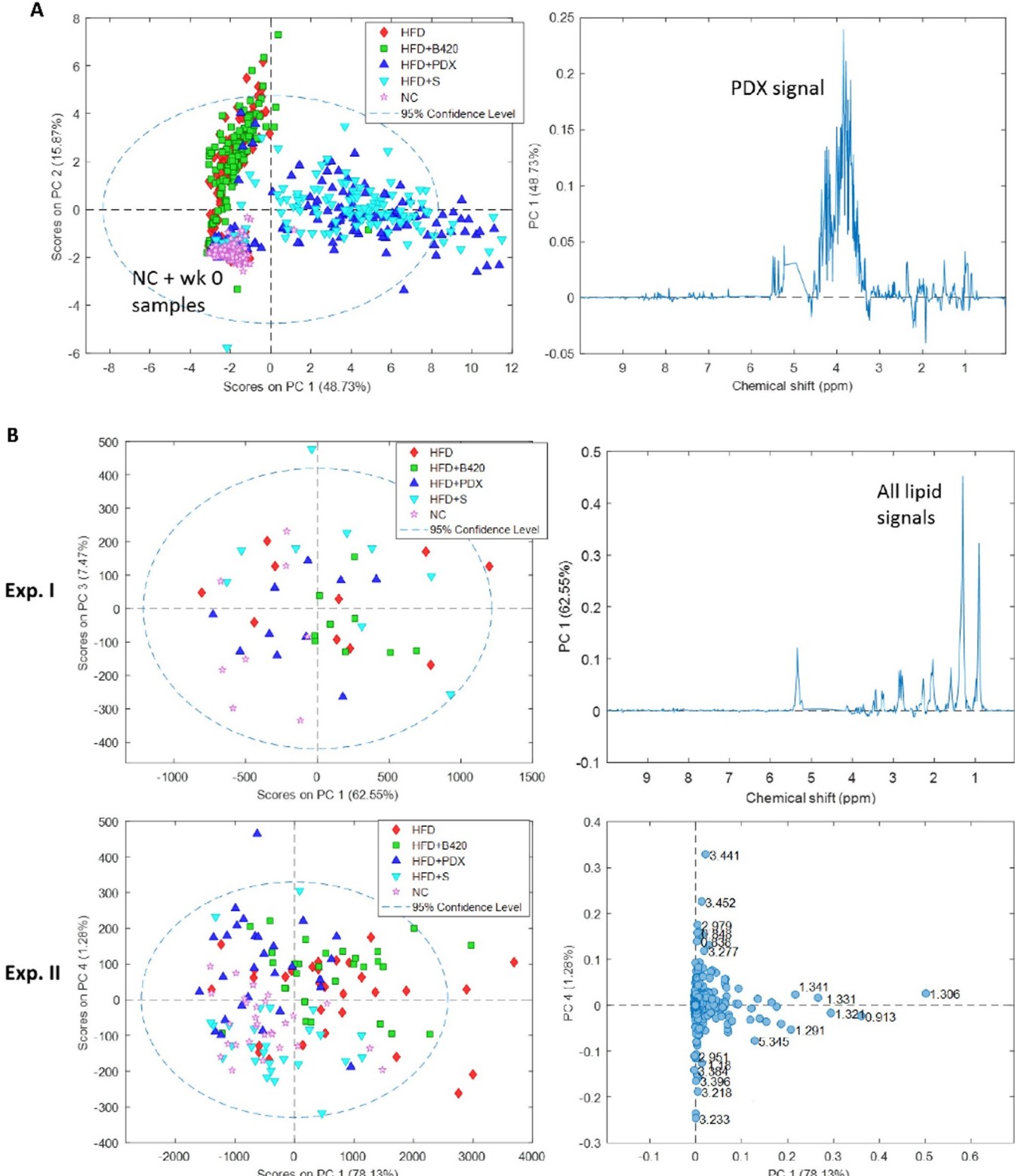

**Fig 3. PCA models performed on the NMR data showing scores and loadings plots.** (A) Fecal samples. (B) intact liver samples.

**Table 1. Mixed models describing treatment and week effects of fecal metabolites.**

| Metabolite | | NC | HFD | HFD+B420 | HFD+PDX | HFD+B420+PDX | P | 3w | 6w | 8w | P | P |
|---|---|---|---|---|---|---|---|---|---|---|---|---|
| | | **Treatment** — LSMeans | | | | | | **Week** — LSMeans | | | | **Treatment* Week** |
| SCFA | Acetate | 0.006[a] | -1.438[c] | -0.919[b] | -0.767[b] | -0.582[b] | <0.0001 | -0.789 | -0.775 | -0.657 | 0.12 | 0.006 |
| | Butyrate | -0.751[ab] | -0.472[ab] | -0.084[a] | -0.984[b] | -0.664[ab] | 0.005 | -0.513 | -0.533 | -0.726 | 0.057 | 0.11 |
| | Formate | 0.807[a] | -0.171[c] | 0.187[bc] | 0.493[ab] | 0.738[a] | <0.0001 | 0.071[b] | 0.493[a] | 0.668[a] | <0.0001 | 0.16 |
| | Isobutyrate | 0.268 | -0.194 | 0.056 | -0.026 | -0.109 | 0.66 | 0.143 | -0.119 | -0.027 | 0.25 | 0.26 |
| | Isovalerate | 0.287[c] | 2.300[a] | 3.004[a] | 1.281[b] | 1.471[b] | <0.0001 | 1.824 | 1.500 | 1.682 | 0.049 | 0.63 |
| | Propionate | -0.196[a] | -1.169[b] | -0.744[ab] | -0.539[ab] | -0.218[a] | <0.0001 | -0.696 | -0.502 | -0.521 | 0.18 | 0.068 |
| | Valerate | -0.285[c] | 1.698[ab] | 2.035[a] | 0.954[b] | 1.420[ab] | <0.0001 | 1.398 | 1.026 | 1.069 | 0.005 | 1.00 |
| BCFA | 2-methyl butyrate | -0.224[b] | 1.484[a] | 1.999[a] | 1.930[a] | 1.993[a] | <0.0001 | 1.576 | 1.286 | 1.449 | 0.12 | 0.45 |
| | 3-methyl-2-oxovalerate | -0.416[b] | 1.757[a] | 2.162[a] | 1.421[a] | 1.768[a] | <0.0001 | 1.265 | 1.312 | 1.439 | 0.28 | 0.074 |
| Alcohols | Ethanol | 0.561[c] | 2.694[a] | 2.688[a] | 1.436[bc] | 1.629[b] | <0.0001 | 1.709[b] | 1.501[b] | 2.194[a] | <0.0001 | <0.0001 |
| | Glycerol | 0.438[b] | 0.403[b] | 0.719[b] | 1.347[a] | 1.469[a] | <0.0001 | 0.751[b] | 0.782[b] | 1.092[a] | 0.001 | 0.74 |
| Amino Acids | Alanine | 0.189[c] | 0.858[b] | 1.456[a] | 1.082[ab] | 1.453[a] | <0.0001 | 0.984 | 0.949 | 1.090 | 0.044 | 0.32 |
| | Arginine | 0.356[a] | -1.744[b] | -1.396[b] | -1.577[b] | -1.635[b] | <0.0001 | -1.388[b] | -0.932[a] | -1.278[b] | <0.0001 | 0.089 |
| | Glycine | -0.001 | 0.013 | 0.006 | 0.014 | -0.003 | 0.65 | 0.005 | 0.003 | 0.009 | 0.57 | 0.23 |
| | Isoleucine | 0.331[c] | 1.520[b] | 2.096[a] | 1.831[ab] | 2.121[a] | <0.0001 | 1.569 | 1.524 | 1.646 | 0.13 | 0.70 |
| | Leucine | 0.270[c] | 1.661[ab] | 2.086[a] | 1.327[b] | 1.594[b] | <0.0001 | 1.438 | 1.303 | 1.422 | 0.15 | 0.59 |
| | Lysine | 0.087[c] | 0.602[b] | 1.005[a] | 0.786[ab] | 1.022[a] | <0.0001 | 0.701 | 0.720 | 0.681 | 0.84 | 0.47 |
| | Phenylalanine | -0.005[c] | 0.612[b] | 1.222[a] | 0.518[b] | 0.713[b] | <0.0001 | 0.316 | 0.851 | 0.668 | 0.44 | 0.57 |
| | Tryptophan | 0.138[c] | 0.422[bc] | 1.061[a] | 0.286[bc] | 0.771[ab] | 0.0001 | 0.606 | 0.444 | 0.557 | 0.19 | 0.11 |
| | Tyrosine | -0.037[c] | 0.719[b] | 1.287[a] | 0.491[b] | 0.729[b] | <0.0001 | 0.463[b] | 0.797[a] | 0.653[ab] | <0.0001 | 0.010 |
| | Valine | 0.062[c] | 1.254[b] | 1.657[a] | 1.172[b] | 1.544[ab] | <0.0001 | 1.119 | 1.110 | 1.184 | 0.50 | 0.98 |
| Metabolism | Fumarate | 0.186[b] | -0.791[c] | -0.562[c] | 1.082[a] | 0.947[a] | <0.0001 | 0.155 | 0.237 | 0.125 | 0.47 | 0.001 |
| | Glutarate | -0.065[c] | 0.775[ab] | 1.204[a] | 0.127[bc] | 0.732[ab] | <0.0001 | 0.674 | 0.492 | 0.499 | 0.17 | 0.33 |
| | Glucose | 0.514[a] | -2.022[d] | -1.440[c] | -0.215[b] | 0.000[ab] | <0.0001 | -0.660 | -0.579 | -0.659 | 0.62 | 0.005 |
| | Lactate | -0.006[a] | -0.904[b] | -0.688[b] | -0.025[a] | 0.203[a] | <0.0001 | -0.316 | -0.313 | -0.224 | 0.37 | 0.10 |
| | Malate | 0.232[d] | 1.568[c] | 1.921[bc] | 2.179[ab] | 2.426[a] | <0.0001 | 1.700 | 1.594 | 1.702 | 0.25 | 0.61 |
| | Ribose | 0.051[bc] | -0.654[d] | -0.465[cd] | 0.803[a] | 0.692[ab] | <0.0001 | 0.100 | 0.123 | 0.033 | 0.79 | 0.77 |
| | Succinate | 0.173[a] | -1.265[c] | -0.956[bc] | -0.190[a] | -0.298[ab] | <0.0001 | -0.472 | -0.530 | -0.519 | 0.81 | 0.016 |
| Others | Cadavarine | -0.329[b] | 1.456[a] | 1.758[a] | 1.113[a] | 1.323[a] | <0.0001 | 1.119 | 0.916 | 1.158 | 0.13 | 0.038 |
| | Orotic acid | -0.582 | -0.865 | -1.066 | -0.589 | -0.216 | 0.14 | -0.920[b] | -0.799[b] | -0.272[a] | <0.0001 | 0.024 |
| | Nicotinate | -0.747 | -0.891 | -0.265 | -1.153 | -0.763 | 0.13 | -0.775 | -0.677 | -0.839 | 0.39 | 0.15 |
| | Trimethylamine | -0.325[c] | 0.004[bc] | 0.500[a] | 0.224[ab] | 0.693[a] | <0.0001 | 0.137 | 0.307 | 0.213 | 0.17 | 0.83 |
| | Xanthine | -0.593 | -0.031 | -0.140 | -0.061 | -0.165 | 0.15 | -0.100 | -0.146 | -0.347 | 0.12 | 0.16 |

The quantified fecal metabolites are log-normal distributed and baseline-adjusted; i.e. a positive and a negative LSMeans value shows increasing and decreasing fecal levels, respectively. Least-square mean values within a row with unlike superscript letters were significantly different for the diet or week effects (P<0.05).

compared to HFD and HFD+B420 groups. In addition, the content of ascorbate (P = 0.029) and glutathione (P = 0.001) in liver were elevated with the inclusion of PDX.

## Spearman correlations

Within group Spearman correlations were tested multi-compartmentally on *Akkermansia* and body weight gain and glucose intolerance (Fig 4A). The correlations between *Akkermansia* in

**Table 2. ANOVA models describing treatment and experiment effects of log-normal distributed quantified liver metabolites values.**

| Metabolite | | NC | HFD | HFD+B420 | HFD+PDX | HFD+B420+PDX | P | I | II | P |
|---|---|---|---|---|---|---|---|---|---|---|
| | | Ln(normalised conc.) | | | | | | Experiment Ln (normalised conc.) | | |
| SCFA & Alcohols | Acetate | -4.79[a] | -5.83[b] | -5.63[b] | -4.73[a] | -4.87[a] | <0.0001 | -4.90 | -5.45 | <0.0001 |
| | Ethanol | -2.74[a] | -3.75[b] | -3.71[b] | -2.82[a] | -3.00[a] | 0.001 | -2.66 | -3.75 | <0.0001 |
| | Glycerol | -4.24[a] | -4.74[b] | -4.85[b] | -4.04[a] | -3.76[a] | <0.0001 | -4.07 | -4.58 | 0.004 |
| Amino Acids | Alanine | -4.83[b] | -5.36[c] | -5.44[c] | -4.28[a] | -4.71[b] | <0.0001 | -4.75 | -5.09 | 0.003 |
| | Glycine | -4.29[a] | -4.93[b] | -4.89[b] | -4.03[a] | -4.25[a] | <0.0001 | -4.35 | -4.61 | 0.06 |
| | Glutamine | -3.67[ab] | -4.20[c] | -3.96[bc] | -3.38[a] | -3.56[ab] | 0.0001 | -3.66 | -3.85 | 0.09 |
| | Isoleucine | -5.24 | -5.92 | -5.94 | -5.56 | -5.75 | 0.58 | -5.53 | -5.83 | 0.13 |
| | Leucine | -3.68 | -4.17 | -4.00 | -3.82 | -3.56 | 0.11 | -3.84 | -3.85 | 0.68 |
| | Valine | -4.67[a] | -5.82[b] | -6.11[b] | -5.13[a] | -5.04[a] | <0.0001 | -5.08 | -5.63 | 0.002 |
| Metabolism & Others | Ascorbate | -4.69[ab] | -4.89[b] | -4.84[b] | -4.29[a] | -4.58[ab] | 0.029 | -4.63 | -4.69 | 0.57 |
| | Betaine | -4.59[bc] | -4.85[cd] | -5.07[d] | -4.09[a] | -4.31[ab] | <0.0001 | -4.30 | -4.86 | <0.0001 |
| | Choline | -4.75[a] | -5.24[b] | -5.35[b] | -4.42[a] | -4.40[a] | <0.0001 | -4.45 | -5.21 | <0.0001 |
| | Creatine | -6.16[ab] | -7.09[c] | -6.98[c] | -5.95[a] | -6.48[b] | <0.0001 | -6.29 | -6.77 | 0.008 |
| | Glucose | -0.48[a] | -0.88[b] | -0.91[b] | -0.29[a] | -0.47[a] | 0.0008 | -0.30 | -0.92 | <0.0001 |
| | Glutathione | -4.01[b] | -4.47[c] | -4.37[bc] | -3.75[a] | -3.90[a] | 0.001 | -4.21 | -3.99 | 0.18 |
| | Glycerophosphocholine | -2.48[b] | -2.59[b] | -2.58[b] | -1.76[a] | -2.05[a] | <0.0001 | -2.24 | -2.35 | 0.25 |
| | Glycogen | 2.52[a] | 2.11[b] | 1.90[b] | 2.57[a] | 2.66[a] | 0.046 | -1.38 | -1.78 | 0.07 |
| | Lactate | -3.70[a] | -4.32[b] | -4.21[b] | -3.56[a] | -3.75[a] | 0.0002 | -3.72 | -4.09 | 0.006 |
| | Taurine | -1.57 | -1.78 | -1.75 | -1.33 | -1.50 | 0.089 | -1.38 | -1.79 | 0.002 |
| | Lipid | 1.68[c] | 2.28[ab] | 2.42[a] | 1.72[c] | 2.02[b] | <0.0001 | 1.89 | 2.16 | 0.004 |

Least-square mean values within a row with unlike superscript letters were significantly different for the treatment or experiment effects (P<0.05).

all compartments and the fecal and liver multi-compartmental metabolite levels were also tested (Fig 4B and 4C).

Intriguingly, *Akkermansia* in adipose tissue was positively correlated with *Akkermansia* in ileum in the HFD+B420 (r = 0.41, P = 0.02) and HFD+PDX (r = 0.41, P = 0.03) groups. In ileum of the HFD+S group, *Akkermansia* was inversely correlated with body weight gain (r = -0.53, P = 0.01) and glucose tolerance (r = -0.33, p = 0.08).

Correlations between fecal metabolites and *Akkermansia* indicated that for HFD+S mice fecal *Akkermansia* was positively correlated with fecal fumarate levels (r = 0.47, P = 0.01) and negatively correlated with fecal propionate (r = -0.55, P<0.01) and nicotinate (r = -0.60, P<0.01). It was also observed in the adipose tissue that the B420 treatment induced an inverse correlation between fecal glycerol and *Akkermansia* (r = -0.40, P = 0.03) and the synbiotic treatment induced an inverse correlation between fecal glycerol and *Akkermansia* (r = -0.50, P<0.01). Ileal *Akkermansia* in HFD+S mice were positively correlated with fecal trimethylamine (TMA; r = 0.40, P = 0.03) and hepatic choline (r = 0.40, P = 0.03) and betaine (r = 0.42, P = 0.02). In the mice fed HFD + B420, a positive correlation between ileum *Akkermansia* and hepatic levels of Taurine (r = 0.44, P = 0.01) and Leucine (r = 0.40, P = 0.03) was seen. In colon, in the case of HFD+PDX fed mice, *Akkermansia* showed inverse correlations to hepatic creatinine (r = -0.40, P = 0.03), choline (r = -0.41, P = 0.03), valine (r = -0.44, P = 0.02) and ethanol (r = -0.47, P = 0.01). Furthermore, synbiotic treatment results in inverse correlations between *Akkermansia* in adipose tissue and hepatic glycine (r = -0.38, P = 0.04), valine (r = -0.39, P = 0.03), choline (r = -0.41, P = 0.02) and positive correlation to hepatic lipid (r = 0.40,

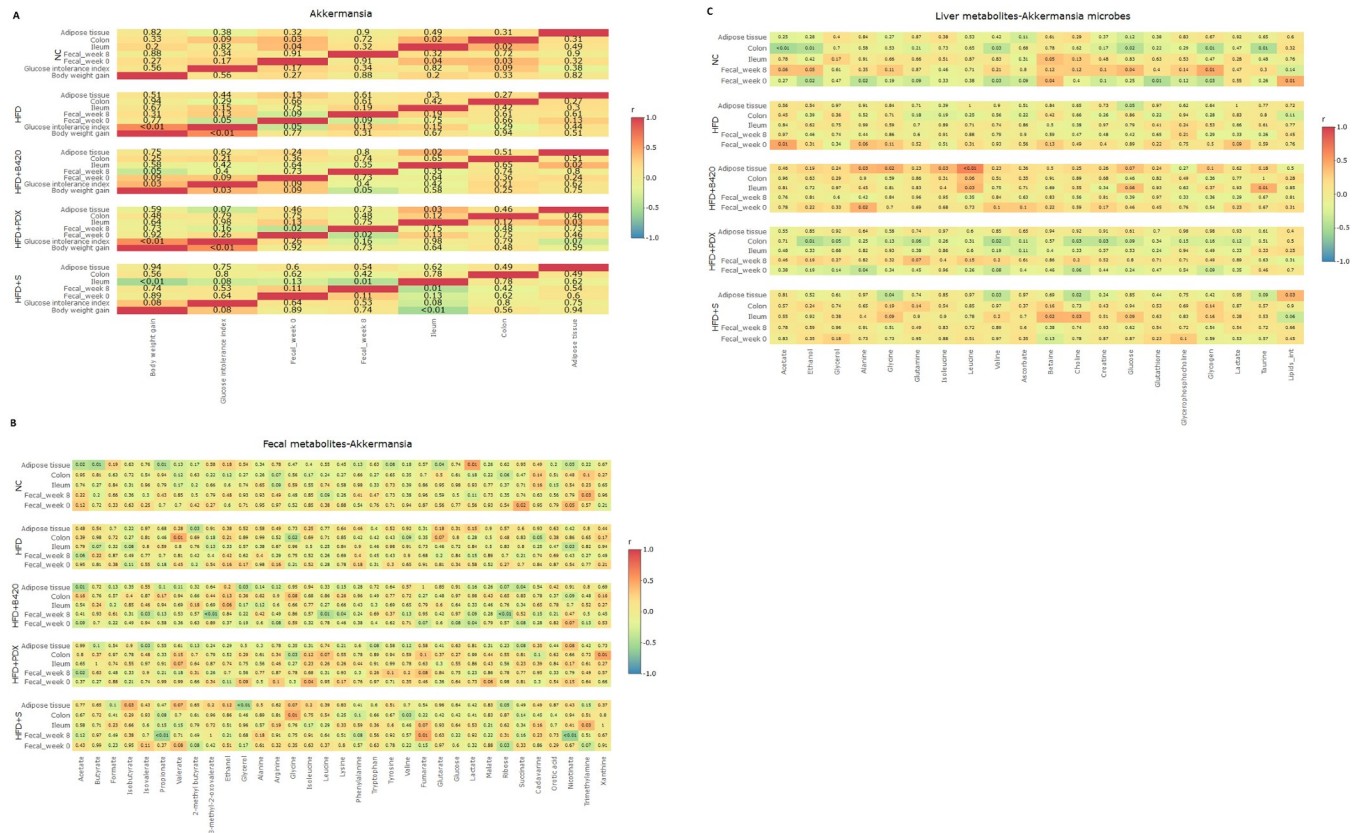

**Fig 4. Spearman's rank correlation coefficients heat map of associations to the relative abundance of *Akkermansia*.** (A) At different compartment, body weight gain and glucose tolerance. (B) Fecal metabolites. (C) Liver metabolites.

P = 0.03). HFD+B420 had positive correlations within group between *Akkermansia* in adipose tissue and liver amino acids such as glycine (r = 0.43, P = 0.02), Alanine (r = 0.40, P = 0.03) as well as branched chain amino acids leucine (r = 0.64, P<0.01) and isoleucine (r = 0.39, P = 0.03). No significant correlations were found for the other treatments in adipose tissue.

## Discussion

In this study, the mechanism of probiotic, prebiotic and synbiotic treatment on fat accumulation was studied by administering HFD (with 60 kcal% fat) to C57BL/6J mouse model susceptible to diet induced obesity. As expected, HFD resulted in drastic weight gain in C57BL/6J mice and induced a dramatic shift in gut microbiota composition by increasing *Firmicutes* and decreasing *Bacteroidetes* in phylum level–this diet-induced reshape of the composition in microbial community under HFD is well-characterized and in accordance with previous literature [41]. This modulation of the *Firmicutes*/*Bacteroidetes*-ratio has been reported to be associated with overweight or obese phenotype not only in animal trials but also in observational human trials both in adults [42] and in children [42]. HFD also decreased the α-diversity. All in all, the fecal microbiota changes induced in the present trial by HFD in C57BL/6J mice were similar to what has previously been described in literature [43].

The primary result of this nine weeks' intervention study was that the metabolic response in C57BL/6J mice to HFD was modified by all three active diets and accompanied by the multi-compartmental changes in microbiota composition and metabolite profiles. It was

expected from previous research that the prebiotic and synbiotic treatments would alleviate the effects of HFD on microbiota in fecal samples and in the gut [7, 8, 11]. However, the present study added to the previously shown ability of PDX to decrease food intake due to its satiety effect [16, 17], an ability to attenuate body weight gain and high hepatic fat levels induced by HFD. On the other hand, pro- and synbiotics did not induce such reduced weight gain in this obesogenic mouse model. Intriguingly, it should be noted that the ability of PDX to attenuate body weight gain demonstrated in this mice HFD study contrasts with what has previously been observed in a human study [11].

An overview of the main findings and on how *Akkermansia* correlates to liver metabolism in a tissue & diet specific manner is illustrated in Fig 5. Importantly, prebiotic PDX both alone and in combination with probiotic B420 increased the prevalence of *Akkermansia* both in gut and in MAT. *Akkermansia muciniphila* has previously been associated with lean phenotype and healthy metabolism [28], possibly via reversing HFD-induced decrease in mucin layer thickness and number of goblet cells in the epithelium [30] by outer membrane protein-Toll-like receptor 2-interaction [31]. Thus, one of the most striking new finding of this study was that synbiotic intervention increased the level of *Akkermansia* in MAT. Utilization of *Akkermansia* as a probiotic is limited by its requirements for complex culture conditions and animal-based medium as well as its sensitivity to oxygen [44]. Although, a recent study by Depommier *et al.* [29] conducted a human clinical trial on oral supplementation of *A. muciniphila*, which was found to be safe in longer term administration. Increasing the relative proportion of *Akkermansia* in the gut microbiota by prebiotics was studied in a clinical trial, in which B420 not only together with PDX but also alone was able to increase the prevalence of *A. muciniphila* in human fecal samples during six-month intervention [25]. Another recent

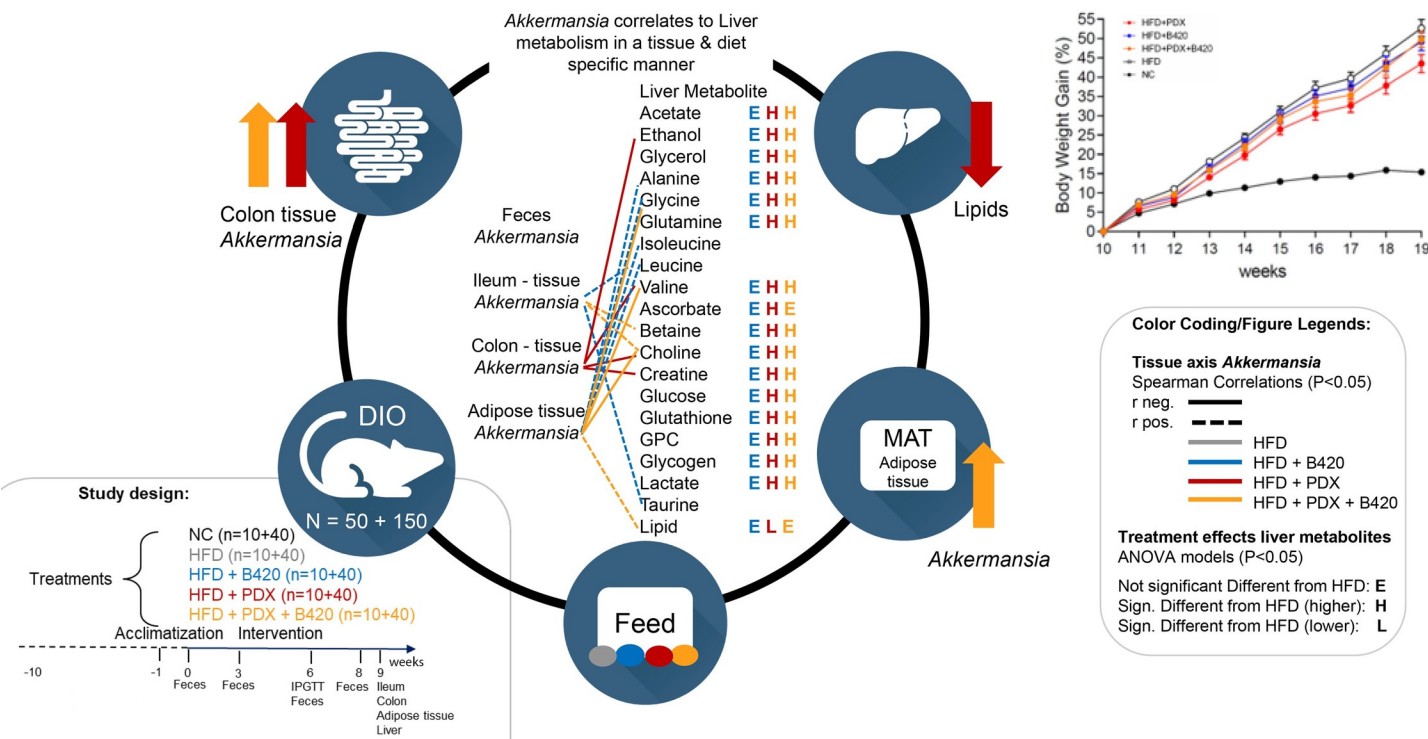

**Fig 5. Graphical abstract of findings including overview of the *Akkermansia* abundance in different tissues with Spearman rank correlations to liver metabolites (color coded to treatment).** The treatment effect of each liver metabolite is given relative to HFD (legends E, H, L) based on the ANOVA model in Table 2.

study by Ondee *et al.* [45] with a similar HFD mice model as the present study, found that Lactobacillus acidophilus 5 (LA5) administration promotes *A. muciniphila* in the gut and showed a clear attenuation of the adverse effects of HFD as indicated by decreased hepatic injury. The present study offers further means to gain mechanistical understanding on the modes of action that pre-, pro- and synbiotics might induce healthy metabolism via increased *Akkermansia* proportion.

Bacterial metabolites are known to exert many of the observed effects that gut microbiota composition has on health. SCFAs form a group of metabolites produced by bacterial fermentation from dietary components such as fiber. In general, associations between fecal levels of SCFAs and obesity is still rather contradictory. Kimura showed that the diet-induced obesity was linked to a decrease in SCFA levels [46] while an increase in levels was observed in genetically modified (*ob/ob*) mice [2]. Further, the mechanisms through which SCFAs can modulate weight gain include appetite regulation. SCFAs have been shown to regulate the G Protein-coupled receptor activity by increasing its expression in adipose tissue and decreasing it in colon [47]. Specifically, propionate has been shown to affect appetite regulation [48] through PYY and GLP-1 hormone release stimulation from human colonic cells. In a clinical trial, B420 alone, as well as together with PDX as a synbiotic product, modulated the dietary intake of the participants so that they consumed less calories than the placebo group, although no dietary recommendations or intervention was given [11]. PDX alone is known to increase the release of GLP-1 postprandially in humans [17, 18]. *A. muciniphila* has been shown to produce propionate, a SCFA with shown effects on genes that regulate for example lipid metabolism and lipolysis [49, 50]. In this study, HFD decreased fecal SCFA levels (acetate, formate and propionate) when compared to normal chow, and synbiotic product with both PDX and B420 was able to statistically significantly increase the SCFA levels compared to HFD. Also, a trend was observed for propionate also in the probiotic and prebiotic groups compared to HFD. Butyrate level was increased in the probiotic group alone compared to all other groups. Thus, the present study showed that PDX and B420 can modulate the gut microbiota under HFD towards composition sustaining propionate production. One can speculate that one of the mechanisms behind the observed ability of PDX and B420 to modulate the calorie intake in clinical trial [11], could be modulation of the appetite hormone level release from human colonic cells by sustained propionate production.

The levels of TCA-cycle metabolites in the fecal samples were increased by PDX, as an increase of fumarate, malate, succinate levels in the HFD+PDX and HFD+S groups compared to HFD and HFD+B420 groups was seen. This kind of metabolic turnover could be expected from the prebiotic effect of the PDX-fiber, and the findings were in accordance with a previous human study in which the intake of PDX was correlated with increased levels of succinate, acetate, butyrate and propionate in fecal metabolome [9].

Gut-liver axis refers to the interaction between gastrointestinal tract and liver which in both healthy and diseased state involves transfer of gut microbiota derived molecules. The close anatomical and functional association between intestinal lumen and liver control the metabolism in the liver tissue. When there is surplus translocation of bacterial components or bacteria towards liver tissue, TLR signaling is modulated and levels of inflammatory markers get elevated [51]. Low choline-diet promotes accumulation of triacylglycerols in the liver, and choline deficiency is also associated to fecal microbiota composition inducing variation in Firmicutes and Proteobacteria species that are directly associated with changes in the liver fat [52]. It is known that HFD may promote distortion of gut-liver axis communication also by inducing alterations in the bile acid and choline metabolism [51]. In the present study, the liver metabolomics showed PDX driven effect of normalized levels of various amino acids, as well as betaine, choline, glycerolphosphocholine, glucose and creatine in the metabolome of

intact liver. Betaine functions in liver as a methyl-donor donor for methionine biosynthesis [53]. Furthermore, betaine is hepatoprotective having i.e. anti-inflammatory effect in liver, and has been associated with reduced liver injury [54]. Choline is an important essential nutrient having various roles in sustaining healthy liver function [55]. Betaine, on the other hand, can be oxidized from choline in a two-step reaction. Choline and its metabolites have crucial roles in maintaining liver health as well as regulating gene expression and seem to even function as a neuroprotectant as reviewed recently by Wiedeman *et al.* [56]. Glutathione levels were observed to be improved by both the PDX and synbiotics intervention, hence improving the detoxification capacity of the liver [57]. Besides being a reactive oxygen species scavenger, human clinical studies indicate that glutathione administration has positive effects on patients with diagnosed fatty acid liver disease [58]. In a study by Gogiashvili *et al.* [59], perturbations in the betaine and transsulfuration pathways have been observed in fatty acid liver disease in *ob/ob* mice by profiling of the liver using HR-MAS NMR. The authors speculate in limitation of glutathione synthesis caused by these pertubations.

As the key finding of this study was to show that *Akkermansia* can be found in mice adipose tissue, and that the infiltration of *Akkermansia* is increased by the synbiotic probiotic and prebiotic intervention, it was very interesting to take a closer look into how *Akkermansia abundance* in various tissues correlates with liver metabolome. *Akkermansia* prevalence in ileum was negatively associated with body weight gain, and positively correlated with Leucine and Taurine for the HFD+B420 group and similar for the hepato-protectants betaine and choline in liver for in the synbiotic group. *Akkermansia* levels in the adipose tissue (MAT) was positively associated ($P<0.05$) with amino acid levels of Alanine, Glycine, Leucine, Isoleucine in the liver of mice fed the HFD + B420 diet. On the contrary, during the HDF + Synbiotic diet a negative correlation was observed between Glycine, Valine and the Choline while hepatic Lipid was negatively associated. Schneeberger et al., 2015 [60] observed a positive correlation of levels of *Akkermansia* to an altering of metabolic parameters related to adipose fat browning and lipid oxidation. Brown fat tissue energy metabolism has recently been associated with changes in the catabolism of branched chained amino acids [61]. It can be hypothezised that the synbiotic diet alters lipid oxidation and brown fat metabolism through the adipose *Akkermansia* infiltration.

A recent paper by Perakakis et al. [62] describes the underlying mechanistic effect of liraglutide and elafibranor, which are a GLP-1 receptor analog and a dual PPAR α/δ agonist, respectively, in regard to treatment of non-alcoholic fatty liver disease (NAFLD). Similarities between the two studies were found for HFD+PDX and HFD+S groups with liraglutide by restoring the levels of glycogen and increasing betaine concentration in the liver. Additionally, both HFD+PDX and HFD+S showed similar results with elafibranor treatment considering the increased glutathione levels in the liver. It has previously been discussed in literature that probiotics supplementation is a potential therapeutic strategy towards NAFDL [63]. Thus, it can be speculated, administration of PDX alone or in combination with B420 could potentially be a supplement to liraglutide and elefibranor to be used in the treatment of NAFLD.

In conclusion, polydextrose and B. animalis lactis ssp. 420, alone and in combination, induced multi-compartmental changes in microbiota and metabolite levels in an obesogenic mouse model. Supplementation of the dietary fiber, polydextrose, was reducing the weight gain during the high fat diet intervention. An increase in the prevalence of *Akkermansia* and improved liver health as indicated by methyl-donors was observed after supplementation of polydextrose with or without probiotics *Bifidobacterium animalis* ssp. *lactis* 420. However, further studies are needed to clarify the effect on liver health incl. e.g. histology and hepatic in-vitro models. The Synbiotic intervention increased the abundance of *Akkermansia* in mice adipose tissue, which again was correlated to reduced weight gain. Understanding the

mechanisms behind gut microbiota association with lean/obese phenotype as well as weight development is highly demanded. The results of the present study bring valuable new information in understanding how pre- and probiotics can affect the interplay between gut, liver and adipose tissue.

## Supporting information

**S1 Fig. Relative abundance at Genus level from metagenomic sequencing.** (A) Fecal samples at week 0. (B) Fecal samples at week 3. The order of the treatments is from left to right: NC, HFD, HFD+B420, HFD+PDX and HFD+S.
(PDF)

**S2 Fig. Microbiota α-diversity index evaluated using the Shannon index.** (A) Fecal samples at week 0. (B) Fecal samples at week 8. (C) Ileum samples. (D) Colon samples. (E) Adipose tissue (MAT).
(PDF)

**S3 Fig. Principal coordinate analysis (PCoA) on microbiota β-diversity clustering based on BrayCurtis, Jaccard, Unifrac and Weighted Unifrac distances for comparison of the different treatments.** (A) Fecal samples at week 0. (B) Fecal samples at week 8. (C) Ileum samples. (D) Colon samples. (E) Adipose tissue (MAT) samples.
(PDF)

**S4 Fig. Representative 1H NMR spectra from the same synbiotic HFD+PDX+B420 mice.** (A) Fecal samples. (B) Intact liver samples.
(PDF)

## Acknowledgments

Anita Beck-Rasmussen and Nina Eggers for technical support with the metabolomics. Ashley Hibberd, Buffy Stahl and Wesley Morovic for performing the PCR and sequencing. The authors also express their gratitude to Arthur Ouwehand for critical review of the manuscript.

## Author Contributions

**Conceptualization:** Sampo Lahtinen, Lotta K. Stenman.

**Data curation:** Christian Clement Yde, Henrik Max Jensen, Niels Christensen, Henna-Maria Kailanto.

**Formal analysis:** Christian Clement Yde, Henrik Max Jensen, Florence Servant, Benjamin Lelouvier.

**Funding acquisition:** Christian Clement Yde.

**Methodology:** Christian Clement Yde.

**Project administration:** Henrik Max Jensen, Kaisa Airaksinen.

**Visualization:** Christian Clement Yde.

**Writing – original draft:** Christian Clement Yde, Henrik Max Jensen, Kaisa Airaksinen, Henna-Maria Kailanto.

**Writing – review & editing:** Christian Clement Yde, Henrik Max Jensen, Niels Christensen, Florence Servant, Benjamin Lelouvier, Sampo Lahtinen, Lotta K. Stenman, Kaisa Airaksinen, Henna-Maria Kailanto.

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
