## [Decision Letter · Decision Letter 0]

25 May 2021

PONE-D-21-07582

Polydextrose with and without Bifidobacterium animalis ssp. lactis 420 drives the prevalence of Akkermansia and improves Liver Health in a Multi-compartmental obesogenic Mice Study.

PLOS ONE

Dear Dr. Yde,

Thank you for submitting your manuscript to PLOS ONE. After careful consideration, we feel that it has merit but does not fully meet PLOS ONE’s publication criteria as it currently stands. Therefore, we invite you to submit a revised version of the manuscript that addresses the points raised during the review process.

We look forward to receiving your revised manuscript.

Kind regards,

Nikolaos Perakakis

Academic Editor

PLOS ONE

Journal Requirements:

3. Please include your tables as part of your main manuscript and remove the individual files. Please note that supplementary tables (should remain/ be uploaded) as separate "supporting information" files

4.Thank you for stating the following in the Financial Disclosure section:

"This study was funded by IFF Health & Biosciences. CCY was financially supported by the Innovation Fund Denmark (Project No. 4228-00010B)."

We note that one or more of the authors are employed by a commercial company: IFF Enabling Technologies and Vaiomer

We also note that one or more of the authors have an affiliation to the commercial funders of this research study: IFF Health & Biosciences

Reviewers' comments:

Reviewer's Responses to Questions

**Comments to the Author**

1. Is the manuscript technically sound, and do the data support the conclusions?

Reviewer #1: Yes

2. Has the statistical analysis been performed appropriately and rigorously? 

Reviewer #1: Yes

3. Have the authors made all data underlying the findings in their manuscript fully available?

Reviewer #1: Yes

4. Is the manuscript presented in an intelligible fashion and written in standard English?

Reviewer #1: Yes

5. Review Comments to the Author

Reviewer #1: I am thankful for the opportunity to review this manuscript. Apparently, I would like to declare that the conceptual design of this work is very similar to what my team and I recently published in Scientific Reports (https://www.nature.com/articles/s41598-021-85449-2). I believe I was selected as a reviewer because of my relevant work. Hence, please allow me to provide comments with regard to it.

The major differences were that the present work investigated the effect of PDX and B. animalis ssp. lactis 420 on a few outcomes and made a conclusion about liver health improvement whereas mine assessed the comprehensive effects of L. acidophilus LA5 on weight reduction, regional fat accumulation, lipidemia, liver injury (liver weight, lipid compositions, and liver enzyme), gut permeability defect, endotoxemia, and serum cytokines; using both in vitro and in vivo evidences.

Interestingly, both B. animalis lactis 420 and L. acidophilus LA5 contributed to the fat reduction through the Akkermansia promotion mechanism. Although at least two probiotics show this potential, many other probiotics may have insufficient evidence so the Short Title and the Conclusions should be more specific. (ie. PDX and B. animalis lactis improves liver health and change levels of Akkermansia in obese mice).

It is essential to demonstrate and/or discuss how the PDX and B. animalis lactis 'move' to liver and other organs. In our L. acidophilus LA5 study, we demonstrated the gut leakage mechanism. Also, potential toxicity, inflammation, and injury of the PDX and B. animalis lactis on the liver should also be included.

Please discuss whether PDX could have a similar or different effect with probiotics other than B. animalis lactis.

The design of this study implicitly suggested the preventive effect of B. animalis lactis 420 but not treatment effect. Please mention this limitation in the Discussion. Also, the improvement of liver health was assessed without in vitro evidence, which should be addressed as another limitation.

6. PLOS authors have the option to publish the peer review history of their article (what does this mean?). If published, this will include your full peer review and any attached files.

Reviewer #1: **Yes: **Krit Pongpirul

---

## [Author Response · Author response to Decision Letter 0]

9 Jul 2021

Thank you for the revision of my manuscript. I have tried to address the concerns raised by you and the reviewer, and I am confident that this has improved the manuscript. Consequently, I hope you will consider the revised manuscript for publication in PLOS ONE, and I am looking forward to be hearing from you. 

Journal Requirements:

Figure citations have been changed to meet PLOS ONE's style requirement. E.g. Fig. 1A has been changed to Fig 1A.

Heading for all major sections have been modified to Level 1 (Bold type, 18pt font).

Multiple figures have been cited as Figs instead of Fig.

The figure captions have been moved after the paragraph in which they are first cited.

2. Please include captions for your Supporting Information files at the end of your manuscript, and update any in-text citations to match accordingly.

“Supporting information” title has been added.

The order of the supporting figures has been changed.

S1_Fig is now provided as a single file.

Supporting figure citations have been changed to meet PLOS ONE's style requirement.

3. Please include your tables as part of your main manuscript and remove the individual files. Please note that supplementary tables (should remain/ be uploaded) as separate "supporting information" files.

Tables have been included after the paragraph in which they are first cited.

Table legend have been moved below the table.

4.Thank you for stating the following in the Financial Disclosure section:

"This study was funded by IFF Health & Biosciences. CCY was financially supported by the Innovation Fund Denmark (Project No. 4228-00010B)."

We note that one or more of the authors are employed by a commercial company: IFF Enabling Technologies and Vaiomer.

We also note that one or more of the authors have an affiliation to the commercial funders of this research study: IFF Health & Biosciences.

As the commercial affiliation did play a role in our study, we have updated the Funding Statement.

Within your Competing Interests Statement, please confirm that this commercial affiliation does not alter your adherence to all PLOS ONE policies on sharing data and materials by including the following statement: "This does not alter our adherence to PLOS ONE policies on sharing data and materials.” (as detailed online in our guide for authors http://journals.plos.org/plosone/s/competinginterests). If this adherence statement is not accurate and there are restrictions on sharing of data and/or materials, please state these.

In the amended Competing Interests Statement, we have declared the commercial affiliation. 

An amended Funding Statement and Competing Interests Statement have been included in the cover letter. 

5. Review Comments to the Author

Reviewer #1: I am thankful for the opportunity to review this manuscript. Apparently, I would like to declare that the conceptual design of this work is very similar to what my team and I recently published in Scientific Reports (https://www.nature.com/articles/s41598-021-85449-2). I believe I was selected as a reviewer because of my relevant work. Hence, please allow me to provide comments with regard to it.

Response: I agree this is a very relevant and interesting paper regarding Akkermansia muciniphilia findings in the present manuscript. 

The major differences were that the present work investigated the effect of PDX and B. animalis ssp. lactis 420 on a few outcomes and made a conclusion about liver health improvement whereas mine assessed the comprehensive effects of L. acidophilus LA5 on weight reduction, regional fat accumulation, lipidemia, liver injury (liver weight, lipid compositions, and liver enzyme), gut permeability defect, endotoxemia, and serum cytokines; using both in vitro and in vivo evidences. 

Response: Both papers have a clear focus on gut microbiota composition after intake of probiotics. The Ondee et al. paper uses as mentioned both in vitro and in vivo methods to investigate the beneficial effects of LA5 by especially gut leakage and liver injury. The present paper is a different approach with a focus on the correlation between the microbiota and the metabolome, which targets the mechanistical understanding on the modes of action of pre- and probiotics intake via increased Akkermansia proportion.

Interestingly, both B. animalis lactis 420 and L. acidophilus LA5 contributed to the fat reduction through the Akkermansia promotion mechanism. Although at least two probiotics show this potential, many other probiotics may have insufficient evidence so the Short Title and the Conclusions should be more specific. (ie. PDX and B. animalis lactis improves liver health and change levels of Akkermansia in obese mice).

Response: This a very important point that both probiotics show the increase in Akkermansia. Therefore, the Ondee et al. paper has been cited in the discussion in the present manuscript.

“Another recent study by Ondee et al. [46] with a similar HFD mice model as the present study, found that Lactobacillus acidophilus 5 (LA5) administration promotes A. muciniphila in the gut and showed a clear attenuation of the adverse effects of HFD as indicated by decreased hepatic injury”.

To be more specific, ‘PDX and B420’ have been used instead of ‘pre, pro and synbiotics’ in the Short Title and the Conclusion. 

It is essential to demonstrate and/or discuss how the PDX and B. animalis lactis 'move' to liver and other organs. In our L. acidophilus LA5 study, we demonstrated the gut leakage mechanism. Also, potential toxicity, inflammation, and injury of the PDX and B. animalis lactis on the liver should also be included.

Response: We found that prebiotic PDX both alone and in combination with probiotic B420 increased the prevalence of Akkermansia in the gastrointestinal tract. Also, the most striking new finding of this study was that synbiotic intervention increased the level of Akkermansia in adipose tissue. In Stenman et al. (EBioMedicine 13 (2016) 190–200; ref. [11]), we found some subtle indications of inflammation and gut leakage (in humans during a 6-month intervention period as an effect of the consumption of PDX and B420. In comparison with the Ondee et al. paper, PDX and B420 interventions show almost no effect on body weight gain (only HFD+PDX was significant compared to HFD). Thus, I would expect the present study to shown little/or no gut leakage mechanism. Of course, it could be of great interest with a study of the effect of PDX and B420 intake on systematic inflammation. However, the main objective of the present study was to examine the underlying mechanisms of PDX and B420 treatments explaining the earlier findings in animal studies and human clinical trials on metabolic health. 

Please discuss whether PDX could have a similar or different effect with probiotics other than B. animalis lactis. 

Response: Magro et al. (Nutrition Journal 2014, 13:75) and Forssten et al. (Microb Ecol Health Dis. 2015; 26) discusses the combination of PDX, L. acidophilus NCFM® and B. lactis HN019 in a yogurt on intestinal transit and PDX, Lactobacillus acidophilus NCFM, and L. paracasei Lpc-37 effects on the growth of C. difficile, respectively. It is a relevant question, but I’m not aware of the combination of PDX and other probiotics than B420 regarding a similar research question as the present study. 

The design of this study implicitly suggested the preventive effect of B. animalis lactis 420 but not treatment effect. Please mention this limitation in the Discussion. Also, the improvement of liver health was assessed without in vitro evidence, which should be addressed as another limitation.

Response: Yes – agree this is a limitation to the study, which would be of great importance to examine in the future. The conclusion has been rewritten to reflect this:” An increase in the prevalence of Akkermansia and improved liver health as indicated by methyl-donors was observed after supplementation of polydextrose with or without probiotics Bifidobacterium animalis ssp. lactis 420. However, further studies are needed to clarify the effect on liver health incl. e.g. histology and hepatic in-vitro models.”

---

## [Decision Letter · Decision Letter 1]

15 Sep 2021

PONE-D-21-07582R1Polydextrose with and without Bifidobacterium animalis ssp. lactis 420 drives the prevalence of Akkermansia and improves Liver Health in a Multi-compartmental obesogenic Mice Study.PLOS ONE

Dear Dr. Christian Clement Yde,

Thank you for submitting your manuscript to PLOS ONE. After careful consideration, we feel that it is an interesting article and has merit but does not fully meet PLOS ONE’s publication criteria as it currently stands. Therefore, we invite you to submit a revised version of the manuscript that addresses the points raised during the review process.

ACADEMIC EDITOR: Thank you for submitting your article in PLOS one. I apologize for the delay which was due to difficulties in the recruitment of reviewers in the summer period. Your manuscript has been carefully reviewed from two reviewers and they both agree that this is a very interesting study and a solid work. Before accepting the manuscript for publication, we would like you though to try to address the remaining comments of the reviewers, especially of reviewer 2. Reviewer 2 has also committed to review the revised form of the manuscript in a timely manner.

Please submit your revised manuscript within 4 weeks. If you will need more time than this to complete your revisions, please reply to this message or contact the journal office at plosone@plos.org. Please include the following items when submitting your revised manuscript:A rebuttal letter that responds to each point raised by the academic editor and reviewer(s). You should upload this letter as a separate file labeled 'Response to Reviewers'.A marked-up copy of your manuscript that highlights changes made to the original version. You should upload this as a separate file labeled 'Revised Manuscript with Track Changes'.An unmarked version of your revised paper without tracked changes. You should upload this as a separate file labeled 'Manuscript'.If applicable, we recommend that you deposit your laboratory protocols in protocols.io to enhance the reproducibility of your results. Protocols.io assigns your protocol its own identifier (DOI) so that it can be cited independently in the future. For instructions see: https://journals.plos.org/plosone/s/submission-guidelines#loc-laboratory-protocols. Additionally, PLOS ONE offers an option for publishing peer-reviewed Lab Protocol articles, which describe protocols hosted on protocols.io. Read more information on sharing protocols at https://plos.org/protocols?utm_medium=editorial-email&utm_source=authorletters&utm_campaign=protocols.

We look forward to receiving your revised manuscript.

Kind regards,

Nikolaos Perakakis

Academic Editor

PLOS ONE

Journal Requirements:

Additional Editor Comments (if provided):

Thank you for submitting your article in PLOS one. I apologize for the delay which was due to difficulties in the recruitment of reviewers in the summer period. Your manuscript has been carefully reviewed from two reviewers and they both agree that this is a very interesting study and a solid work. Before accepting the manuscript for publication, we would like you though to try to address the remaining comments of the reviewers, especially of reviewer 2. Reviewer 2 has also committed to review the revised form of the manuscript in a timely manner.

Reviewers' comments:

Reviewer's Responses to Questions

**Comments to the Author**

1. If the authors have adequately addressed your comments raised in a previous round of review and you feel that this manuscript is now acceptable for publication, you may indicate that here to bypass the “Comments to the Author” section, enter your conflict of interest statement in the “Confidential to Editor” section, and submit your "Accept" recommendation.

Reviewer #1: All comments have been addressed

Reviewer #2: (No Response)

2. Is the manuscript technically sound, and do the data support the conclusions?

Reviewer #1: Yes

Reviewer #2: Yes

3. Has the statistical analysis been performed appropriately and rigorously? 

Reviewer #1: Yes

Reviewer #2: Yes

4. Have the authors made all data underlying the findings in their manuscript fully available?

Reviewer #1: Yes

Reviewer #2: Yes

5. Is the manuscript presented in an intelligible fashion and written in standard English?

Reviewer #1: Yes

Reviewer #2: Yes

6. Review Comments to the Author

Reviewer #1: Thank you very much for the revised manuscript. All of my comments have been addressed satisfactorily.

Nonetheless, there are some minor formatting issues to be corrected. For example, the 'author' of the first reference should be 'OECD', not '2019 O'.

Reviewer #2: Clement Yde et al. present an impressive assessment of murine metabolomics and metagenomics following five (4+control) different treatment combinations of prebiotic and probiotic elements alongside a high-fat diet. The study is well-written and conveys an interesting message, whereas its methods section comprehensively describes all implemented modalities. The authors have provided extensive evaluations of murine microbiota both at treatment initiation and study endpoint, showcasing relative similarities in diversity indices upon initiation, which is important given the numerous relevant confounding factors (cage, bedding, basal animal conditions etc.) The metabolomic arm could be ideally supplemented with the inclusion of biopsy images from tissues obtained and the analysis and discussion of liver histology and/or immunohistochemistry pertaining to non-alcoholic fatty liver disease.

In its current form the manuscript is well delineated and the reviewer believes the authors should follow the suggestions below. However, if available, the inclusion of data on murine body composition and plasma concentrations of insulin, liver enzymes, triglycerides, conventionally-measured cholesterol fractions etc would be of great scientific interest, since it could ascertain the precise pathobiological fingerprint of all described interventions that reflect the altered metabolomic profiles and their correlation with microbiota.

Additionally, some further suggestions are in order to improve overall readability and quality.

• Consider describing the full dietary energy composition of the murine HFD (total calories, %carbs, %protein, %fat)

• Was there any attrition / mice deaths before study completion?

• Gut microbiome constitutes a primary driver of hepatic steatosis and inflammation, both components of non-alcoholic fatty liver disease, whereas relevant interventions through probiotic administration are being discussed across the literature as potential therapeutic strategies (Meroni et al., Nutrients . 2019 Nov 4;11(11):2642; Delzenne et al., Nat Med 2018; 24: 906–907; Le Roy et al., Gut 2013; 62(12): 1787-94) Consider expanding your discussion and thus further abridging the two main areas of your study (microbiota + metabolomics) towards this direction, and to address the functional significance and potential long-term clinical relevance of your findings regarding liver metabolomics, especially considering weight gain differences between human and murine models as described in your already published work, already cited (Ref. 11), and the similarities or differences of metabolomic findings observed herein to recently published evidence on murine NAFLD therapeutics regarding liraglutide (especially relevant owing to the known PDX-induced increase of GLP-1) /elafibranor/empagliflozin administration in HFD-fed mice with HFD-induced non-acoholic steatohepatitis (Perakakis et al., Liver Intl 2021 Aug;41(8):1853-1866).

• Consider providing an additional image, embedded in Figure 5, in the form of a time scale graph/diagram showcasing the study design, randomization and number of total mice per group at study end, and measurements per week

• Figure 1: Consider including all between-group significances in a small table included within the image, rather than in the figure legend

Minor:

• If possible, consider increasing the DPI of your tiff figures to ensure that all legends are easily readable in their final form, because owing to the small fonts, it can be hard for a curious reader to discern them, especially in the components of figure 2 (A-E)

• Optionally, consider including supplementary figures alongside their respective legends in a separate PDF file to facilitate readability and continuity rather than including captions separately and uploading a zip file with images only;

• Consider carefully screening the text for minor English language mistakes, especially adverb utilization, tense discrepancies etc.

7. PLOS authors have the option to publish the peer review history of their article (what does this mean?). If published, this will include your full peer review and any attached files.

Reviewer #1: **Yes: **Assoc. Prof. Dr. Krit Pongpirul, MD, MPH, PhD.

Reviewer #2: No

---

## [Author Response · Author response to Decision Letter 1]

2 Nov 2021

[Author] Thank you again for the second revision of our manuscript and providing valuable input for improving the manuscript. Below, we have addressed the remaining comments and suggestions of the reviewers.

Journal Requirements:

[Author] No cited papers have been retracted.

Reviewer's Responses to Questions

Comments to the Author

6. Review Comments to the Author

Reviewer #1: Thank you very much for the revised manuscript. All of my comments have been addressed satisfactorily.

Nonetheless, there are some minor formatting issues to be corrected. For example, the 'author' of the first reference should be 'OECD', not '2019 O'.

[Author] Thank you, this has been amended accordingly.

Reviewer #2: Clement Yde et al. present an impressive assessment of murine metabolomics and metagenomics following five (4+control) different treatment combinations of prebiotic and probiotic elements alongside a high-fat diet. The study is well-written and conveys an interesting message, whereas its methods section comprehensively describes all implemented modalities. The authors have provided extensive evaluations of murine microbiota both at treatment initiation and study endpoint, showcasing relative similarities in diversity indices upon initiation, which is important given the numerous relevant confounding factors (cage, bedding, basal animal conditions etc.) The metabolomic arm could be ideally supplemented with the inclusion of biopsy images from tissues obtained and the analysis and discussion of liver histology and/or immunohistochemistry pertaining to non-alcoholic fatty liver disease.

[Author] I fully agree, it would be of great relevance to incl. liver histology in the study. It has been previously been discussed in the response to the first review of the manuscript. In the discussion, it has been stated as an important focus area for future research: “However, further studies are needed to clarify the effect on liver health incl. e.g. histology and hepatic in-vitro models”.

In its current form the manuscript is well delineated and the reviewer believes the authors should follow the suggestions below. However, if available, the inclusion of data on murine body composition and plasma concentrations of insulin, liver enzymes, triglycerides, conventionally-measured cholesterol fractions etc would be of great scientific interest, since it could ascertain the precise pathobiological fingerprint of all described interventions that reflect the altered metabolomic profiles and their correlation with microbiota.

[Author] In the clinical trial described in the Stenman et al. paper (EBioMedicine 13 (2016) 190–200; ref. [11]), we report serum insulin, ASAT, ALAT, gamma-glutamyltransferase, total cholesterol, LDL, HDL and triglycerides etc.. Unfortunately, these standard clinical analyses were not incl. in the present study as the main objective was to examine the underlying mechanisms of PDX and B420 treatments explaining the earlier findings in animal studies and human clinical trials on metabolic health. 

Additionally, some further suggestions are in order to improve overall readability and quality.

• Consider describing the full dietary energy composition of the murine HFD (total calories, %carbs, %protein, %fat)

[Author]The caloric information of protein, carbohydrate and energy density have been added to the ‘Animals and study design’ section.

• Was there any attrition / mice deaths before study completion?

[Author]One mouse died at intervention week 7 prior to completion of the study. This has been added to the ‘Animals and study design’ section.

• Gut microbiome constitutes a primary driver of hepatic steatosis and inflammation, both components of non-alcoholic fatty liver disease, whereas relevant interventions through probiotic administration are being discussed across the literature as potential therapeutic strategies (Meroni et al., Nutrients . 2019 Nov 4;11(11):2642; Delzenne et al., Nat Med 2018; 24: 906–907; Le Roy et al., Gut 2013; 62(12): 1787-94) Consider expanding your discussion and thus further abridging the two main areas of your study (microbiota + metabolomics) towards this direction, and to address the functional significance and potential long-term clinical relevance of your findings regarding liver metabolomics, especially considering weight gain differences between human and murine models as described in your already published work, already cited (Ref. 11), and the similarities or differences of metabolomic findings observed herein to recently published evidence on murine NAFLD therapeutics regarding liraglutide (especially relevant owing to the known PDX-induced increase of GLP-1) /elafibranor/empagliflozin administration in HFD-fed mice with HFD-induced non-acoholic steatohepatitis (Perakakis et al., Liver Intl 2021 Aug;41(8):1853-1866).

[Author] Thanks for providing very relevant and interesting literature regarding probiotics potential as a therapeutic strategy towards NAFDL. Indeed, I agree that it is relevant to discuss the similarities with effects of liraglutide and elafibranor.

Added to the discussion: “A recent paper by Perakakis et. al [63] describes the underlying mechanistic effect of liraglutide and elafibranor, which are a GLP-1 receptor analog and a dual PPAR α/δ agonist, respectively, in regard to treatment of non-alcoholic fatty liver disease (NAFLD). Striking similarities between the two studies were found for HFD+PDX and HFD+S groups with liraglutide by restoring the levels of glycogen and increasing betaine concentration in the liver. Additionally, both HFD+PDX and HFD+S showed similar results with elafibranor treatment considering the increased glutathione levels in the liver. It has previously been discussed in literature that probiotics supplementation is a potential therapeutic strategy towards NAFDL [64]. Thus, administration of PDX alone or in combination with B420 could potentially be a supplement to liraglutide and elefibranor to be used in the treatment of NAFLD.”

• Consider providing an additional image, embedded in Figure 5, in the form of a time scale graph/diagram showcasing the study design, randomization and number of total mice per group at study end, and measurements per week

[Author] To enhance the overview, the study design has been included in Fig. 5 as suggested.

• Figure 1: Consider including all between-group significances in a small table included within the image, rather than in the figure legend

[Author] To avoid presenting a too crowded figure, we prefer having the significances in the figure legend. 

Minor:

• If possible, consider increasing the DPI of your tiff figures to ensure that all legends are easily readable in their final form, because owing to the small fonts, it can be hard for a curious reader to discern them, especially in the components of figure 2 (A-E)

[Author] The resolution of the legends in Fig. 2 have been increased for better readability. Furthermore, the font size has been increased for the taxonomy names labelling.

• Optionally, consider including supplementary figures alongside their respective legends in a separate PDF file to facilitate readability and continuity rather than including captions separately and uploading a zip file with images only;

[Author] Thank you for your suggestion, this has been amended accordingly.

• Consider carefully screening the text for minor English language mistakes, especially adverb utilization, tense discrepancies etc.

[Author] Minor language mistakes, mainly tense discrepancies, have been corrected throughout the manuscript.

[Author] Additional changes:

• The fecal samples were collected after 8-weeks but it was a 9-weeks intervention study. In the manuscript, the study was described as a 8-weeks intervention, which has been corrected to 9-weeks. 

• Description of the harvesting of blood in the ‘Sample Collection and Processing’ section has been deleted as the blood analysis was not incl. in the manuscript.

---

## [Decision Letter · Decision Letter 2]

17 Nov 2021

Polydextrose with and without Bifidobacterium animalis ssp. lactis 420 drives the prevalence of Akkermansia and improves Liver Health in a Multi-compartmental obesogenic Mice Study.

PONE-D-21-07582R2

Dear Dr. Clement Yde,

We’re pleased to inform you that your manuscript has been judged scientifically suitable for publication and will be formally accepted for publication once it meets all outstanding technical requirements.

Kind regards,

Nikolaos Perakakis

Academic Editor

PLOS ONE

Additional Editor Comments (optional):

Reviewers' comments:

Reviewer's Responses to Questions

**Comments to the Author**

1. If the authors have adequately addressed your comments raised in a previous round of review and you feel that this manuscript is now acceptable for publication, you may indicate that here to bypass the “Comments to the Author” section, enter your conflict of interest statement in the “Confidential to Editor” section, and submit your "Accept" recommendation.

Reviewer #2: All comments have been addressed

2. Is the manuscript technically sound, and do the data support the conclusions?

Reviewer #2: Yes

3. Has the statistical analysis been performed appropriately and rigorously? 

Reviewer #2: Yes

4. Have the authors made all data underlying the findings in their manuscript fully available?

Reviewer #2: Yes

5. Is the manuscript presented in an intelligible fashion and written in standard English?

Reviewer #2: Yes

6. Review Comments to the Author

Reviewer #2: (No Response)

7. PLOS authors have the option to publish the peer review history of their article (what does this mean?). If published, this will include your full peer review and any attached files.

Reviewer #2: No

---

## [Editor Report · Acceptance letter]

22 Nov 2021

PONE-D-21-07582R2 

Polydextrose with and without *Bifidobacterium animalis* ssp. lactis 420 drives the prevalence of *Akkermansia* and improves Liver Health in a Multi-compartmental obesogenic Mice Study. 

Dear Dr. Yde:

I'm pleased to inform you that your manuscript has been deemed suitable for publication in PLOS ONE. Congratulations! Your manuscript is now with our production department. 

Kind regards, 

on behalf of

Dr. Nikolaos Perakakis 

Academic Editor

PLOS ONE